# Direct observation of coherent energy transfer in nonlinear micromechanical oscillators

Changyao Chen[1], Damián H. Zanette[2], David A. Czaplewski[1], Steven Shaw[3] & Daniel López[1]

Energy dissipation is an unavoidable phenomenon of physical systems that are directly coupled to an external environmental bath. In an oscillatory system, it leads to the decay of the oscillation amplitude. In situations where stable oscillations are required, the energy dissipated by the vibrations is usually compensated by replenishment from external energy sources. Consequently, if the external energy supply is removed, the amplitude of oscillations start to decay immediately, since there is no means to restitute the energy dissipated. Here, we demonstrate a novel dissipation engineering strategy that can support stable oscillations without supplying external energy to compensate losses. The fundamental intrinsic mechanism of resonant mode coupling is used to redistribute and store mechanical energy among vibrational modes and coherently transfer it back to the principal mode when the external excitation is off. To experimentally demonstrate this phenomenon, we exploit the nonlinear dynamic response of microelectromechanical oscillators to couple two different vibrational modes through an internal resonance.

[1] Center for Nanoscale Materials, Argonne National Laboratory, 9700 South Cass Avenue, Argonne, Illinois 60439, USA. [2] Centro Atómico Bariloche and Instituto Balseiro, Comisión Nacional de Energía Atómica. Consejo Nacional de Investigaciones Científicas y Técnicas. 8400 San Carlos de Bariloche, Río Negro, Argentina. [3] Department of Mechanical and Aerospace Engineering, Florida Institute of Technology, 150 West University Bolevard., Melbourne, Florida 32901, USA. Correspondence and requests for materials should be addressed to D.L. (email: dlopez@anl.gov).

Micro and nano-scale mechanical resonators are examples of oscillatory systems that have been studied for decades, since they offer great flexibility to design their mechanical response and their intrinsic and extrinsic dissipation mechanisms[1–3]. For instance, expeditious energy dissipation is required in applications such as vibration isolation or switching, where mechanical motions need to be quickly damped out[4–6]. Conversely, small dissipation rates—or, equivalently, efficient isolation of the system of interest from its environment—are highly desired in other applications because they directly provide more stable frequency sources[7], enhanced detection of extremely weak forces[8,9], and allow for room-temperature quantum-coherent operations and sensing[10–12]. The goal of dissipation engineering is to control the dissipative processes in which the energy stored in the system can be guided towards the natural environment or another resonant mode in a predefined manner. If the interaction with the other modes—such as an electromagnetic cavity[13,14], a light field[15], a phononic crystal[16] or a mechanical vibration[17]—is stronger than the interaction with the thermal bath, a new regime emerges as the system dynamics can be designed to enable distinctive behaviours. Most commonly, parametric modulation is used to activate and control the interaction, where an external modal control is applied to mediate the dynamics. Although parametric modulation can couple any two modes without any prerequisite relationship of their frequencies[13,17], hence capable of bridging distinct domains (for example, optical modes in THz range, electrical modes in GHz range and mechanical below MHz), the disparate frequencies typically make it difficult to achieve strong intermodal coupling.

Here, we introduce a general mechanism that naturally couples mechanical modes with an intermodal coupling rate that greatly exceeds the intrinsic relaxation rate of the coupled modes. This is achieved by coupling different vibrational modes of a single oscillator through an internal resonance. Internal resonance[18] (IR) capitalizes on the condition where the resonant frequencies of two distinct modes satisfy a commensurate relationship to enable strong coupling and efficient energy transfer[19]. Operating micromechanical oscillators in the nonlinear regime, where the resonant frequency has a strong dependence on the oscillation amplitude, allows for precise tuning of the principal mode oscillation frequency just by increasing the applied driving force and thus it can be used to access the IR condition. At IR, energy initially imparted to the principal mode can be continuously exchanged between all the resonantly coupled modes and, if the external energy supply is turned off, this energy can be coherently redirected toward the principal mode, effectively keeping the principal mode of the resonator oscillating. Although IR holds great promise in areas such as frequency stabilization[20] and energy harvesting[21,22], it has never been utilized to engineer dissipation processes in micro and nano-resonators, partly because of the lack of definitive experimental results revealing the details of the energy transfer mechanism.

## Results

**Steady-state characterizations**. Clamped–clamped (c–c) beam resonators are the most common and popular type used in microelectromechanical (MEMS) and nanoelectromechanical (NEMS) devices: they are straightforward to fabricate at the micro and nano-scale, their fundamental dissipation processes have been thoroughly studied[3,23,24], and their nonlinear vibrational response can be precisely tailored[25]. As a consequence, it is possible to fabricate c–c beam resonators with a large detuning range[26] that facilitates the coupling between mechanical modes. In our study, we used a single crystalline silicon c–c beam to demonstrate the IR phenomenon. The natural frequency of the principal in-plane flexural mode, $f_{\text{in-plane}}$, is found to be ~61.4 kHz through electrostatic actuation and detection, with a linear damping rate $\gamma_1/2\pi \approx 0.50$ Hz, corresponding to a quality factor ($Q$) of ~122,680 (Fig. 1a). When the in-plane motion is driven with increased actuation, the large transverse amplitude leads to non-negligible elongation of the beam, causing the restoring force to vary nonlinearly with the displacement[26]. This well-known Duffing nonlinearity shifts the resonant frequency upward with increasing actuation, as shown in Fig. 1b. In this case, the resonant frequency ($f_{\text{res}}$) corresponds to the value where the vibrational amplitude peaks as the drive frequency is swept up. For a Duffing nonlinear resonator, the increase of the resonant frequency scales quadratically with the actuation, or equivalently, with $\nu_{\text{a.c.}}$ in our set-up (Methods section), as is shown in Fig. 1c. The resonant frequency increases quadratically with the driving strength until it reaches $f_{\text{IR}} \approx 64.9$ kHz, where it saturates due to the coupling of the principal mode with a higher frequency mode through an IR. This saturation frequency matches one third of a higher frequency torsional mode of the same c–c beam, whose thermomechanical noise spectrum (measured separately) is shown in Fig. 1d. The natural frequency of this torsional mode is $f_{\text{torsional}} \approx 194.6$ kHz, with a linear damping rate $\gamma_2/2\pi \approx 2.25$ Hz (corresponding to a $Q$ of ~86,505). At this tuning condition, the higher frequency (torsional) mode drains mechanical energy from the principal mode via an inter-modal coupling mechanism: instead of building a larger flexural in-plane mode, the energy input at frequency $f_{\text{IR}}$ is used to excite the torsional mode at a frequency three times larger than $f_{\text{IR}}$ (1:3 mode coupling)[20].

**Transient characterizations**. To better understand the mechanism responsible for energy exchange and dynamic evolution of the IR, we put the resonator in a stationary oscillatory state through a closed-loop configuration, and then perform time-resolved measurements of its oscillation amplitude and frequency before and after the driving force is switched off (ringdown). We apply different self-sustaining schemes that allow the in-plane motion to reach autonomous oscillations whose frequency is determined by the mechanical resonance, which can be further tuned by the feedback force through either a phase-locked loop (PLL)[20] or a feedback phase-delay[27,28]. When the resonator's frequency is outside IR (that is, in-plane oscillation frequency different from 64.9 kHz), the energy decay mechanism is typical of a single-mode nonlinear oscillator[29]: after the excitation is turned off, the amplitude immediately starts to decay towards zero, and the frequency decays toward the natural frequency of the principal in-plane flexural mode, $f_{\text{in-plane}}$ (Fig. 2a,b). The amplitude decay is well approximated by a simple exponential decay, with a dissipation rate of ~0.68 Hz (Fig. 2a, inset). This dissipation rate is larger than the value obtained from quasi-static frequency sweep (Fig. 1a) due to other nonlinear processes dominant at large amplitudes[29] and frequency fluctuations[2]. The temporal evolution of the instantaneous frequency during ringdown, obtained by performing fast Fourier transform on the time-domain data, (Fig. 2b), clearly shows that the instantaneous frequency also decays from its initial value ($\approx 63.6$ kHz) towards the natural frequency of the principal in-plane flexural mode ($\approx 61.4$ kHz). The decay is also approximately exponential, with a decay rate of ~1.39 Hz, that is, twice that of the amplitude, as expected for a Duffing nonlinearity (Supplementary Note 2). When the system is brought into IR as the in-plane oscillation frequency approaches 64.9 kHz, we find an unexpected and qualitatively different ringdown behaviour: after the external excitation is

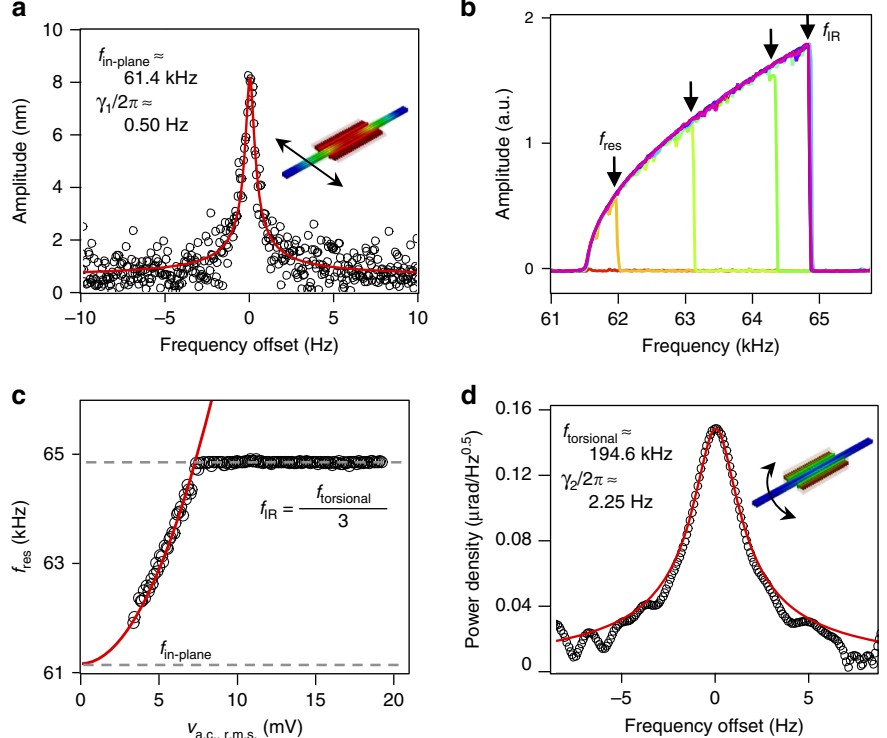

**Figure 1 | Internal resonance in a nonlinear MEMS oscillator. (a)** Linear resonance of the in-plane oscillation mode measured with open-loop set-up, d.c. bias $V_{d.c.} = 7$ V, and alternating current (a.c.) voltage actuation $v_{a.c.} = 40$ μV (r.m.s.). The black circles are acquired data, and the red curve is the corresponding Lorentzian fit. Inset: finite element simulation of the in-plane mode shape (CoventorWare). The length and thickness of the c–c beam are 500 and 10 μm, respectively. The central portion consists of three parallel beams, each with width of 3 μm. **(b)** The vibrational amplitude of the in-plane motion as the external excitation frequency is swept up (open-loop), with excitation amplitude $v_{a.c.}$ increasing from 1 to 19 mV at 2 mV steps. At large $v_{a.c.}$, the vibrational amplitude spectrum shows an asymmetrical line-shape that skews towards high frequencies, due to the positive Duffing nonlinearity. As $v_{a.c.}$ increases, the frequency at which the vibrational amplitude peaks, $f_{res}$, also increases, until reaching $\approx 64.9$ kHz. **(c)** Measured $f_{res}$ (black circles) with different $v_{a.c.}$. For $v_{a.c.} \lesssim 7.5$ mV, $f_{res}$ follows a quadratic dependence (red curve) on $v_{a.c.}$. For $v_{a.c.} \gtrsim 7.5$ mV, $f_{res}$ stabilizes around 64.9 kHz, which equals to one third of $f_{torsional}$. Notice that all the voltages are r.m.s. values. **(d)** Thermomechanical noise spectrum of the torsional mode measured with optical interferometry. The black circles are acquired data, and the red curve is the corresponding Lorentzian fit.

removed, the amplitude and frequency of the in-plane motion remain constant for a certain period of time $t_{coherent}$ (Fig. 2c,d). During $t_{coherent}$, the resonator continues oscillating with a stable sinusoidal waveform as if the external energy supply were still on (Fig. 2e). In other words, it behaves as an ideal autonomous oscillator that requires neither the external sustaining feedback circuitry for power supply nor a frequency reference. Beyond $t_{coherent}$, the principal mode begins the expected exponential decay, both in terms of amplitude and frequency, as in a single-mode nonlinear oscillator[29]. Figure 2f shows the dependence of $t_{coherent}$ on the actuation force, $v_{a.c.}$ in our set-up, suggesting a practical method to control the period of time where the in-plane mode does not decay.

## Discussion

A theoretical description of the results described in Fig. 2 can be obtained with a simplified model with a 1:1 IR condition considering two coupled damped oscillators: a nonlinear Duffing oscillator coupled with a linear one. In this case, the equations of motion for the spatial degrees of freedom, $x_1$ and $x_2$ are expressed as:

$$\ddot{x}_1 + \gamma_1 \dot{x}_1 + x_1 + \beta x_1^3 = Jx_2 + F_{fb},$$
$$\ddot{x}_2 + \gamma_2 \dot{x}_2 + \Omega_2^2 x_2 = J'x_1, \qquad (1)$$

where $\gamma_1$, $\gamma_2$ are the respective dissipation rates, $\beta$ is the Duffing nonlinear coefficient, and $\Omega_2 \gtrsim 1$ is the normalized frequency of

the uncoupled linear oscillator $x_2$. The two oscillators are coupled to each other, with coupling coefficients $J$ and $J'$. Only the nonlinear oscillator is driven by an external feedback force $F_{fb}$, to compensate the energy dissipation and achieve self-sustaining oscillations. For a positive Duffing coefficient, $\beta > 0$, the frequency of the nonlinear oscillator, $\Omega_1$, can be increased by increasing $F_{fb}$. Without coupling ($J = J' = 0$), the system reduces to two independent oscillators, with $x_2$ asymptotically at rest, and $x_1$ behaving as a self-sustaining nonlinear oscillator with tunable frequency[28] (Supplementary Note 1). For finite coupling, the linear oscillator $x_2$ can now be excited due to the motion of the driven nonlinear oscillator $x_1$. When the IR condition is achieved, the linear oscillator is driven resonantly with strength determined by the coupling $J'$. The above analysis, using spatial degrees of freedom, is equivalent to a model describing the natural vibration modes of the system in which the coupling is nonlinear (Supplementary Note 6).

The stationary solutions to equation (1) determine the possible initial conditions of the ringdown evolution which, in turn, is governed by the same equations but with $F_{fb} = 0$. For the case where both oscillators have small dissipation rates ($\gamma_1, \gamma_2 \ll 1$), we can apply a perturbation method to solve equation (1), for both the steady state and the transient response during ringdown[27,30] (Supplementary Notes 3 and 4). Figure 3 shows the numerical solutions of equation (1) for small dissipation ($\gamma_1, \gamma_2 \sim 10^{-5}$), sufficiently large Duffing nonlinear coefficients ensuring that the amplitude of the oscillation is well within the

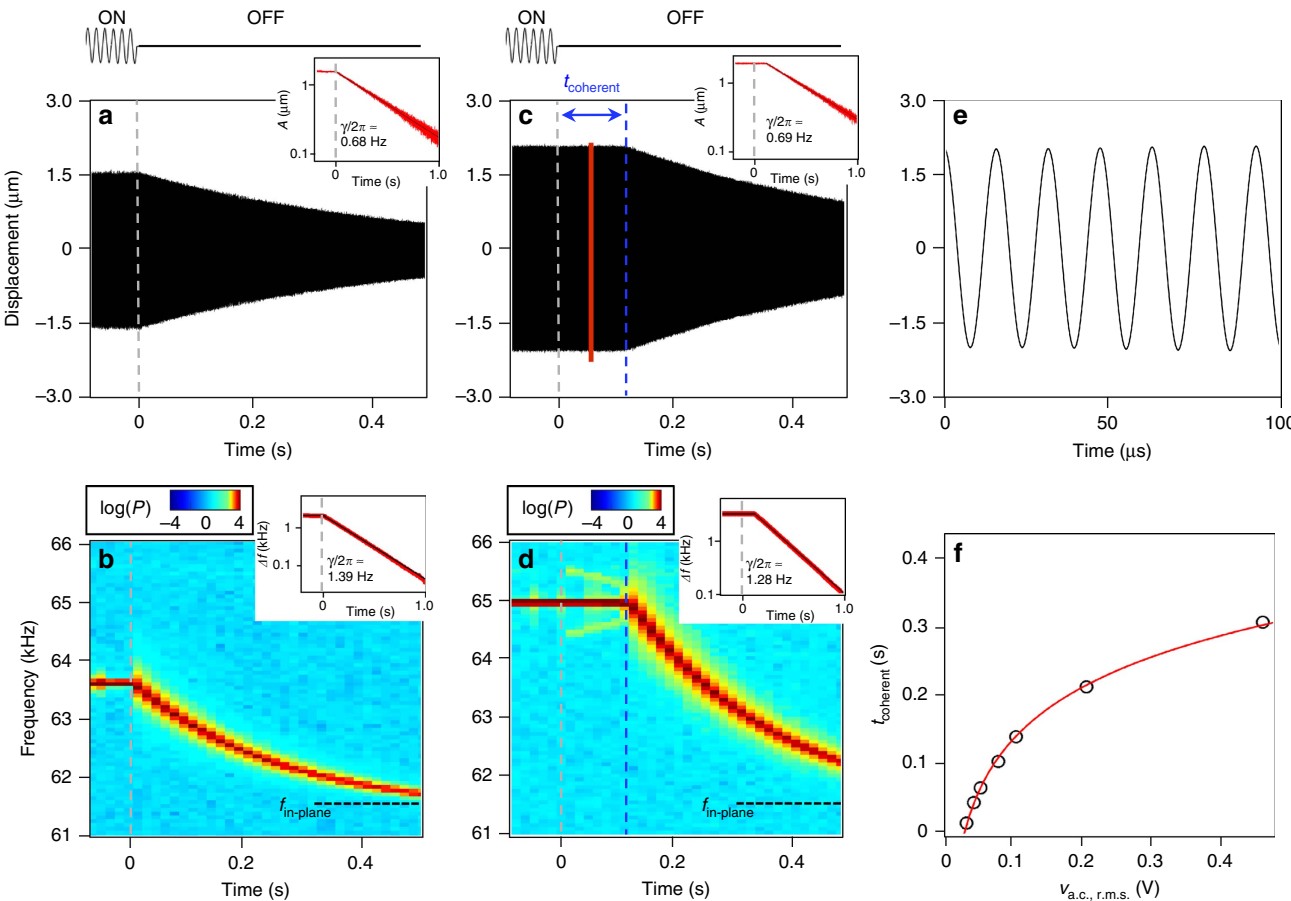

**Figure 2 | Ringdown responses outside and inside internal resonance.** (**a**) Oscillation displacement of the in-plane mode before and after the external drive is turned off (at time equals 0 s), when its oscillation frequency is outside the IR condition. The inset shows the extracted envelope of the displacement, plotted in logarithmic scale. The black line is a fit with an exponential decay, with a decay rate of $\sim 0.68$ Hz. The external excitation $v_{a.c.}$ is 6.3 mV. (**b**) Temporal frequency response of the oscillation during the same period of time. The power spectrum at each nominal time $t_i$ is obtained by performing a non-overlapping fast Fourier transform of the time-domain data in a narrow window of 16 ms centred around $t_i$. The inset shows the extracted instantaneous frequency offset from the principal in-plane frequency, with fitting to an exponential decay (black line). The fitted decay rate is $\sim 1.39$ Hz. (**c**) Displacement of the in-plane oscillations before and after the external drive is turned off, when its oscillation frequency is inside the IR condition. For the first $\sim 108$ ms after the ringdown starts, the envelope of the oscillation remains practically constant. The inset shows the extracted oscillation envelop during ringdown, with an exponential decay fit for the $t > t_{coherent}$ portion. The fitted decay rate is $\sim 0.69$ Hz. (**d**) Temporal response of the oscillation frequency during the same period of time, showing a constant frequency during $t_{coherent}$. The inset shows the extracted instantaneous frequency offset from the principal in-plane frequency, with fitting to an exponential decay (black line) for the $t > t_{coherent}$ portion. The fitted decay rate is $\sim 1.28$ Hz. For $t < 0$, the self-sustaining motion is driven by an external PLL that outputs a single-frequency signal. For $t \in (0, t_{coherent})$ the transient response is only controlled by the system dynamics and two sidebands flanking the main frequency peak appear (Supplementary Notes 4 and 5; Supplementary Fig. 7). (**e**) Zoomed-in view of the oscillation of ringdown of **c**, indicating clean and stable sinusoidal oscillations (the time is offset to 50 ms). (**f**) $t_{coherent}$ (black circles) obtained with different steady-state drive $v_{a.c.}$. The error bars are of the same size or smaller than the symbols, therefore they are omitted here. The red line is a theoretical fit with the model described in equation (1) (Supplementary Note 4).

nonlinear regime ($\beta \sim 10^{-3}$ to $10^{-2}$), and large coupling coefficients ($J \times J' \sim 10^{-9}$ to $10^{-7}$). The time-resolved energy decay of the nonlinear oscillator, $x_1$, is plotted in Fig. 3a, and is compared with its decay outside IR. The simplified model presented above qualitatively reproduces the experimental results of Fig. 2: at IR and for a finite period of time, $t_{coherent}$, the oscillator continues oscillating with practically constant amplitude (Fig. 3a) and frequency (Fig. 3b) after the external energy supply has been switched off. Figure 3c shows the accompanying time evolution of the higher frequency linear oscillator, indicating a rapid, non-exponential decay in its amplitude reaching toward zero at time $t_{coherent}$. These results indicate a net transfer of energy from the higher frequency linear oscillator to the principal one, since the former decays at a rate much faster than the exponential decay rate characteristic of energy dissipation toward the environmental thermal bath (dotted line in Fig. 3c). The difference between the two curves in Fig. 3c provides a direct estimation of the amount of energy transferred during the period $t_{coherent}$. It clearly shows that a significant amount of the energy gets redirected toward $x_1$ instead of being dissipated to the environmental bath. For times longer than $t_{coherent}$, the amplitude of the higher frequency oscillator approaches zero, and the principal one begins its exponential decay toward equilibrium. For the particular case of our experiment, with a 1:3 mode coupling between a predominantly flexural in-plane (lower frequency) mode and a predominantly torsional (higher frequency) mode, the higher frequency mode acts as an energy reservoir for the flexural in-plane (principal) mode, by storing mechanical energy during the stationary state and transferring it to the low frequency mode when the external feedback force is switched off, until its energy is exhausted.

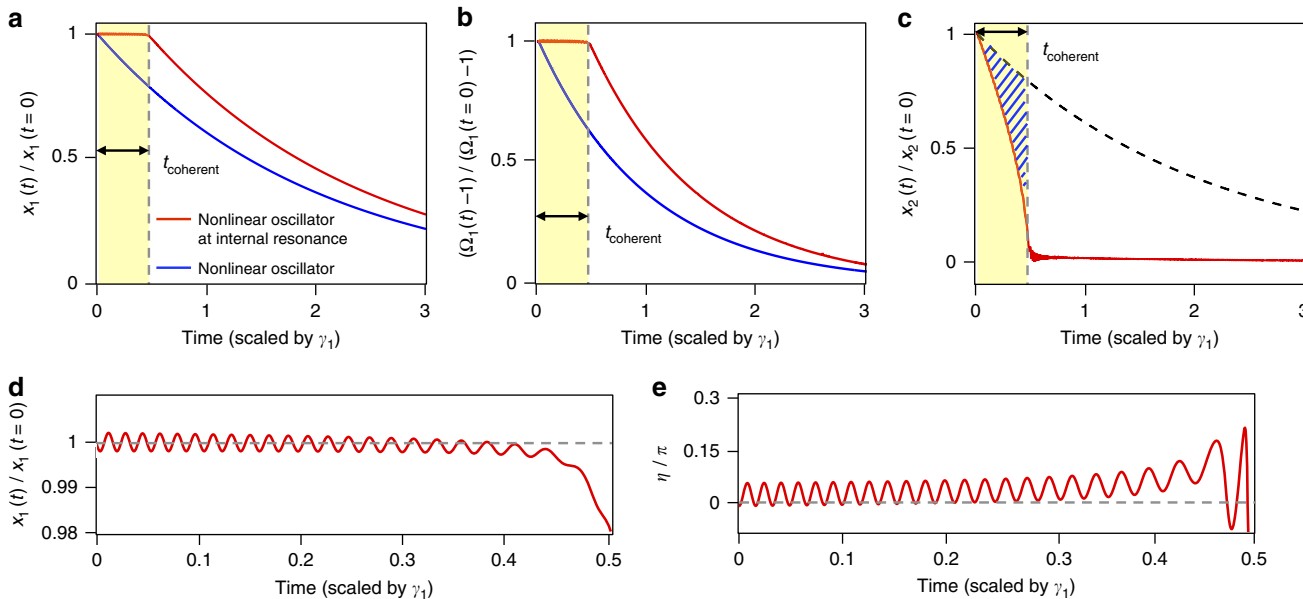

**Figure 3 | Modelling the ringdown response of a nonlinear oscillator.** (**a**) Simulated transient responses of the vibrational amplitude decay (that is, envelope of the oscillation displacement) of a single-nonlinear Duffing oscillator (blue) and a nonlinear oscillator at internal resonance (red). Note that for the nonlinear oscillator at IR, the vibrational amplitude does not decay during $t_{coherent}$ after the external excitation is removed at $t = 0$. (**b**) Simulated transient responses of the instantaneous frequency offset, $\Omega_1 - 1$, for the two cases presented in **a**. After the external excitation is removed, the instantaneous frequency offset for the single-nonlinear oscillator (blue) decays exponentially towards zero, while for the nonlinear oscillator at IR (red), the instantaneous frequency stays constant until $t_{coherent}$. (**c**) Temporal evolution during ringdown of the higher frequency linear oscillator ($x_2$) at IR: the amplitude (red line) decays much faster than the expected exponential decay (dashed black line), indicating faster energy transfer to $x_1$ than to the environmental bath. The shaded area indicates the amount of energy transferred from $x_2$ to $x_1$. (**d**) Time dependence of the amplitude of the principal oscillator ($x_1$) during the period $t_{coherent}$, revealing small oscillations ($\sim 0.1\%$) around its initial value, with a decreasing frequency. (**e**) Simulated phase difference $\eta$ between the two coupled oscillators during the same period of time as in **d**. It shows the presence of coherent oscillations around a positive average value of $\eta$, indicating a net energy transfer from $x_2$ to $x_1$.

A closer look at the time dependence of $x_1$ during $t_{coherent}$ shows coherent oscillations around its steady-state value, with a relatively very small amplitude ($\sim 0.1\%$) and a slowly decreasing frequency (Fig. 3d). Furthermore, the phase difference between the two oscillators, $\eta$, whose value is constant before ringdown, presents similar oscillations that indicates a oscillating flow of energy between the two oscillators (Fig. 3e). The simulated transient responses of $\eta$ shows that it oscillates around a positive value, which implies that, despite the instantaneous direction of energy flow, there is a net energy flow from $x_2$ to $x_1$. Consequently, the amplitude of $x_2$ decays rather abruptly, while the amplitude of $x_1$, remains—on the average—practically constant. This dynamical exchange breaks down as soon as the linear oscillator exhausts its energy, and from then on, the principal oscillator behaves as a single-mode nonlinear oscillator.

While our experimental set-up cannot directly measure the coherent amplitude oscillations shown in Fig. 3d due to the limitation of the measurement bandwidth, we do see their presence in the time-resolved frequency spectrum before and during $t_{coherent}$. Figure 4 shows the temporal evolution of the resonator's frequency as it transits from steady-state to ringdown in a device that is driven to a strong IR condition with larger feedback force $F_{fb}$. During steady-state (regime I) we find two sidebands flanked around the main peak in the power spectrum. During $t_{coherent}$ (regime II), both sidebands evolve towards the frequency of the principal in-plane flexural mode, merging with it at $t = t_{coherent}$. When the frequency, $\Omega_1$, is exponentially decaying (regime III), there is no evidence of sidebands because there is no coupling between the vibrational modes and thus no exchange of energy. The emergence of these sidebands is a direct consequence

of the energy exchange (cf. Fig. 3e), whose value quantifies the exchange rate, $\gamma_{ex}$. For the particular case presented in Fig. 4a, we obtained a $\gamma_{ex} \sim 800\,Hz$, which is almost three orders of magnitude larger than the intrinsic damping rate of each of the coupled modes ($\sim 1\,Hz$). By performing similar frequency measurements at different driving forces, a linear correlation between $t_{coherent}$ and $\gamma_{ex}$ is obtained (Fig. 4a, right inset). The data indicates that a $\gamma_{ex} > 200\,Hz$ is needed to have a finite $t_{coherent}$ and the larger the $\gamma_{ex}$, the larger the $t_{coherent}$.

The experimental and numerical results discussed above demonstrate that mode coupling can be used to engineer the intrinsic relaxation phenomena of nonlinear oscillators. In particular, when different vibrational modes are coupled through an IR, the exchange of energy between modes could happen orders of magnitude faster than the exchange of energy with the external environmental bath (Fig. 4b). Under these conditions, nonlinear resonators can sustain, for a finite period of time, stable oscillations without external energy supply. The dissipation engineering concept presented in this work could be applied to a wide range of MEMS and NEMS oscillators whose performance is limited by the electrical noise in the feedback circuit[31]. MEMS and NEMS resonators oscillating without external power should be ideal devices to identify the ultimate stability limit imposed by thermomechanical noise[32]. The possibility to control the energy exchange rate between coupled modes creates a testbed to validate theories of thermalization of nonlinear systems out of equilibrium[33,34] and anomalous friction in nonlinear resonators[4,35]. In atomically thin NEMS resonators, where the nonlinear dynamic response can be easily achieved[36], mode coupling can have a significant effect on the relaxation process toward equilibrium[37].

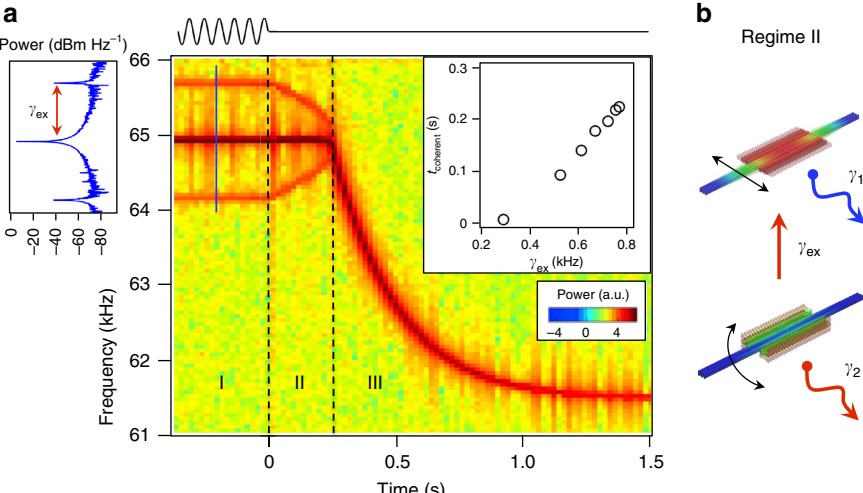

**Figure 4 | Time scale of the energy exchange dynamic at internal resonance. (a)** Temporal evolution of the instantaneous frequency before (regime I) and during (regime II and III) ringdown. For $t < 0$, the self-sustaining motion is driven by a linearly feedback signal with appropriate phase-delay, instead of using a PLL set-up as the one used to obtain the data presented in Fig. 2. In regime I, there are two sidebands flanking the main steady-state frequency (64.9 kHz, IR) that indicate the energy exchange rate between the main and higher frequency modes (a single-power spectrum is shown in left inset). During $t_{coherent}$ (regime II), the main frequency stays constant and the sidebands merge toward the main frequency peak, and finally disappear at the end of regime II. Regime III represents the normal decay for a nonlinear Duffing oscillator and thus shows no evidence of sidebands. Right inset: extracted $\gamma_{ex}$ plotted against corresponding $t_{coherent}$, showing a linear correlation between them. The error bars are smaller than the size of the symbols. **(b)** Schematic representation of the energy flow during $t_{coherent}$ (regime II): there is a net energy flow from the high frequency torsional mode to the in-plane principal mode at a rate $\gamma_{ex} \gg \gamma_1, \gamma_2$. This large difference in relaxation rates causes the in-plane principal mode to maintain stable oscillations even after the external power supply has been switched off.

It is worthwhile emphasizing the manifold role of nonlinearity in the occurrence of the phenomena studied here. In the first place, nonlinearity is responsible for the coupling between different oscillation modes, enabling the exchange of energy that determines their mutual influence. At the same time, the upshift of the oscillation frequency as the driving force increases in amplitude makes it possible that the nonlinear oscillator reaches the condition of IR—an effect which is absent in harmonic oscillators. Finally, the presence of an interval of coherence just after the driving force is switched off has also to be ascribed to nonlinearity. The combination of beating and exponential decay that characterizes the dynamics of coupled linear oscillators is in fact not able to account for such temporary stability of the oscillation amplitude in the absence of an external action.

## Methods
**Device fabrication and measurements.** The MEMS devices used here are fabricated by MEMSCAP with standard SOIMUMPs process (http://www.mems-cap.com/products/mumps/soimumps). The device is placed in a vacuum chamber, with pressure about $3 \times 10^{-5}$ Torr. The capacitive comb drive and sensing scheme is designed to actuate and detect the in-plane motion of the c–c beam. With a d.c. bias $V_{d.c.}$ applied to the beam, and an a.c. voltage $v_{a.c.}$ applied to one of the comb electrode (drive electrode), the a.c. excitation force along the in-plane direction, $F_{ex}$, between the c–c beam and the stationary electrodes is:

$$F_{ex} = \frac{C_d'}{2}(V_{d.c.} + v_{a.c.})^2 \approx C_d' V_{d.c.} v_{a.c.}, \qquad (2)$$

where $C_d'$ is the spatial derivative of the capacitance (between c–c bean and drive electrode) along the in-plane direction, and we also drop the d.c. term of $V_{d.c.}^2$. Therefore, for fixed d.c. bias $V_{d.c.}$, the excitation scales linearly with $v_{a.c.}$. To characterize the in-plane mode, we applied a small $v_{a.c.}$ to the drive electrode (with Switch 1 set to position 2), and measured the response as capacitive current $i(\omega)$ from the other comb electrode (sense electrode), with a lock-in amplifier (Zurich Instrument HF2LI). Given the total capacitance between the c–c beam and the sense electrode of $C_s = 8.85$ fF (calculated from the device geometry), and assuming a sinusoidal motion of $x = A \sin(\omega t)$, the measured root mean square (r.m.s.) value of the amplitude $A$ at frequency $\omega$ can be expressed as $A(\omega) = (i(\omega)g)/(\sqrt{2}\omega N \varepsilon_0 t)$, where $g = 2 \mu m$ is the gap between the opposing comb fingers, $N$ is the total number of fingers, $\varepsilon_0$ is the vacuum permittivity, and $t = 10 \mu m$ is the thickness of the finger. In this manner, the measured electrical signal is converted to mechanical

displacement, and we have calibrated such conversion[28]. In practice, a voltage (instead of current) is measured due to the employment of trans-impedance amplifier (FEMTO DLPCA-200). A typical data is shown in Fig. 1a.

To monitor other vibration motions, such as out-of-plane and torsional, we employ an optical interferometry method, with the laser spot focused on the outer-most position of the movable structure, where both the out-of-plane mode and torsional mode can be detected as out-of-plane motion. Therefore, both the out-of-plane and torsional modes can be detected. The interference signal is then amplified and recorded with either a digital oscilloscope or a spectrum analyzer. The low noise level and superior sensitivity of the optical transduction allow us to directly resolve the mechanical resonance driven merely by thermal noise, as shown in Supplementary Fig. 2. We then extract the resonant frequencies and the (linear) quality factors from the power spectra. The responsivity of the interferometry set-up is separately calibrated with a reference mirror oscillating with large amplitude (larger than the wavelength of the laser, 635 nm), and such responsivity is later used to convert the measured thermal noise power spectrum density to the data shown in Fig. 1d. The values of the measured mechanical resonances of 159.83 and 194.64 kHz agree well with the values of the out-of-plane and the torsional modes that are predicted from finite element simulations (CoventorWare), which are 186.60 and 211.94 kHz, respectively. The discrepancies are likely due to the residual stress from the fabrication process. However, we do not observe IR between the out-of-plane mode and the in-plane mode in our experiments.

Since the laser spot size ($\sim 4.5 \mu m$) is comparable to the largest lateral dimension ($< 6 \mu m$) of the c–c beam, when the in-plane motion is set into self-sustained oscillation, the intensity of the optical interference is also modulated at the in-plane oscillation frequency and its harmonics, as the c–c beam is oscillating across the laser spot. Supplementary Fig. 3 shows the measured optical interference power spectrum with the device operated in self-sustained motion, with the in-plane oscillation frequency of 63.2 kHz. Both the third harmonic of the in-plane mode (189.6 kHz) and the torsional mode (194.6 kHz, driven by thermal noise) are present. However, at IR the third harmonic of the in-plane mode and the torsional mode coincide, therefore we can not distinguish their individual response with optical interferometry.

## Self-sustaining oscillation and ringdown around internal resonance.
When both switches in Supplementary Fig. 1 are set to position 1, we engage a closed-loop configuration to enable self-sustaining motion of the in-plane mode. The oscillation is initiated by thermal fluctuation, and the stability of the oscillation is ensured by either a PLL inserted in the feedback circuitry, or without a PLL but through elastic nonlinearity of the c–c beam itself[28]. In both cases, we can adjust the effective feedback force to tune the in-plane oscillation frequency. In the former case, the excitation force that feeds back to drive the oscillator is directly regulated

by controlling the output amplitude of the PLL signal ($v_{a.c.}$), whereas in the latter case, the effective excitation force is controlled by adjusting the phase-delay induced by the feedback circuitry. We observe similar frequency detuning and identical IR with both schemes (Supplementary Fig. 4).

After the in-plane oscillation has reach steady-state, the ringdown is triggered by turning Switch 2 from position 1 to 2. We do not find any discontinuity in the oscillation waveform before and after the switching action (Supplementary Fig. 5). The delay from the switch (Mini-Circuits ZASWA-2-50DR+) is <100 ns.

**Data availability.** The data supporting the findings of this study are available upon request.

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

## Acknowledgements

We would like to thank Mark Dykman, Jeffrey Guest, Aftab Ahmed, Ori Shoshani, Pavel Polunin, Thomas Kenny, Adrian Bachtold, Andreas Isacsson and Vladimir Aksyuk for critical discussions. Use of the Center for Nanoscale Materials at the Argonne National Laboratory was supported by the U.S. Department of Energy, Office of Science, Office of Basic Energy Sciences, under Contract No. DE-AC02-06CH11357. S.S. is supported by NSF grant 1561829 and funds from Florida Institute of Technology.

## Author contributions

D.L. conceived the experiments, C.C. and D.C. performed the experiments, D.H.Z., S.S. and C.C. performed the modelling and numerical simulations, C.C. and D.L. co-wrote the paper. All authors analysed the data, discussed the results and commented on the manuscript.

## Additional information

**Competing interests:** The authors declare no competing financial interests.

