## [Peer Review File · Nature Communications]

Reviewers' comments:

Reviewer #1 (Remarks to the Author):

The ms presents experimental observations of energy transfer between different "modes" in a nonlinear micro mechanical oscillator. The authors propose a description of the phenomenon in terms of nonlinear dynamics by introducing a Duffing oscillator coupled to a linear one.

The results are interesting. The experimental observation is novel and the phenomenon has potential applications in cases of practical interests.

The experimental results are accurate and convincing, although a direct observation of the amplitude change of the high-frequency mode at the IR (Internal Resonance) condition is missing due to the limitations of the readout (optical) system employed by the authors.

The narrative of the paper is somehow misleading (probably beyond the author's intentions). In fact this is presented as a way to avoid mechanical energy dissipation, while it is just a way to prolong self sustained oscillations by extracting energy from a different mode. Nevertheless a remarkable phenomenon. A careful rephrase of the introductory part of the ms could take care of the problem.

A brief mention of the role of noise in the energy transfer process could widen the scope of the paper and potentially increase its interest.

In conclusion I find the work sound and potentially interesting for a wide audience. Worth publishing.

Reviewer #2 (Remarks to the Author):

The paper by Chen et al. details ring-down experiments made on a MEMS resonator displaying a 3:1 internal resonance (IR). Their main finding is that free decay from initial conditions near the IR show a period of sustained coherent oscillations of the fundamental mode after the driving is switched off. They reproduce qualitatively the behavior using a simple model of linearly coupled oscillators.

The article is rather well written and for the most part clear, and deserves publication in some form. I have one major comment however (broken up into several below), that I think must be addressed before the article can be considered for publication.

1. The two modes which are in resonance are in a 3:1 frequency ratio. As the authors point out very clearly on at least two places in the article, an internal resonance between the modes require a nonlinear coupling. Indeed, a coupling term in the Hamiltonian of the form $H_{12} = x_1^2 x_2$ would for instance be able to do this.

1a) In the model they present, a linear coupling is used ($H_{12} = x_1 x_2$). Can one argue that the excellent agreement between theory and experiment comes from a linearization of the nonlinear coupling at the amplitude near the IR? If so, this should be explained more clearly in the text.

1b) Towards the end of the supplementary, the authors argue that the coupling between the modes is linear. Indeed, it is explicitly written in the end of the paragraph after Eqs (S19a,b) that "..., therefore causing a linear coupling between the oscillations...". I do not understand how this linear coupling can account for the observed 3:1 internal resonance. This should be explained.

1c) While the linearized coupling facilitates analytical estimates, I see no hindrance to simulating

numerically the system with a nonlinear coupling for a true 3:1 internal resonance. I suggest the authors do so also in the supplementary in conjunction with figure S.10. This would be more convincing.

1d) In all the modeling, all numbers are given in a dimensionless form. What is the relation between these numbers to actual physical values? (I am looking for actual numbers here). Can they be determined independently? Do the same parameters that reproduce the ringdown also reproduce the driven response? If one estimates the coupling constants J, J' do they agree with the ones in the simulations? How sensitive is the main result (stable amplitude of mode 1 after switching off the drive) to choice of parameters, i.e. is it a robust phenomenon or does it require careful parameter tuning?

1e) In relation to figure 4, the quantity γ_{ex} is introduced as the exchange rate. How does γ_{ex} relate to J, J' ? It seems it depends on drive power, why?

Reviewer #3 (Remarks to the Author):

The result presented here is nice and simple: basically the authors have shown that the secondary mode of a resonator with 1:3 internal resonance transfers energy to the principal mode with a much faster rate than which of the considered modes, so that the principal mode sustains constant amplitude oscillations rather than decaying exponentially like a linear resonator.

The experimental data looks very sound, results are clearly explained and well supported. This builds on a large body of work on inter-modal energy transfer mechanisms produced by the non-linear dynamics community in the last 50 years.

I feel the authors might want to explain better why this is both surprising and important, starting with the title: aren't most of known energy transfers (parametric, sub/superharmonic, internal...) "coherent" and "directly observed"?

The authors write that concerning internal resonances, there is a "lack of definitive experimental results revealing the details of the energy transfer mechanism". I am not sure what they mean by that exactly. Could the authors relate this to both prior knowledge and their own result? If they have the sidebands (fig 4) in mind, hasn't this beating phenomenon already been shown by, for example, the work of Vakakis and co-authors (eg Nonlinear Targeted Energy Transfer in Mechanical and Structural Systems, Effect of 1:3 resonance on the steady-state dynamics of a forced strongly nonlinear oscillator with a linear light attachment AAM2014), or that of Manevitch and co-authors (ref23)? There is also much work performed by Pernot et al. .

Also, how does the results here compare to the work of Nayfeh and Mook, non-linear oscillations and in particular section 6.3.2 and figure 6-11 where both principal and secondary decay? (in this case, both modes have third order NL and the coupling is nonlinear too, can the authors comment on their linear coupling model?).

More importantly, I am not sure I understand the potential impact of the result. In the authors' minds, how can this be used to "enable a new generation of autonomous devices"? Timescales of interest here are a few 100ms. They write this can be applied to oscillators limited by "electrical noise in the feedback circuit". How so? What do they mean by "identifying the ultimate stability imposed by thermomechanical noise"? My understanding is that frequency fluctuations discussed in Ref36 do not originate in the external power source. As for non-equilibrium systems, I believe theories mostly refer to non-linear coupling (like coupling between flexural modes, eg Barnard et al, PNAS2012, Venstra et

al APL2012), as opposed to the linear coupling considered here.

Again, I believe the result presented here is very nice, clearly shown and provides insight into the mechanism ; I am just not sure the impact is sufficient for Nature Communications.

Reviewer #4 (Remarks to the Author):

It has been a pleasure reviewing the paper entitled "Direct observation of coherent energy transfer in nonlinear micro-mechanical oscillators" by Chen et al. In recent years, continuing developments in micro and nanoelectromechanical systems (MEMS/NEMS) has fueled a renaissance in the field of miniaturized mechanical sensors and actuators. These devices have attained a great deal of consideration primarily since the practical significance of tailoring specific dynamical properties opens endless possibilities in the construction of a new class of sensors, actuators and new materials. Furthermore, MEMS and NEMS systems have produced an enormous impact in disparate scientific disciplines ranging from engineering to physical and life sciences, and furthermore have many commercial manifestations. Over the past few decades, these systems have shown to represent an ideal platform interface between the classical and quantum domains.

Nonlinear micromechanical systems represent structures that exhibit rich dynamical phenomena, and further have exquisite sensitivity near instabilities and allow for efficient configurational control of using external fields. The authors describe coherent energy transfer mechanisms in nonlinear micro-mechanical oscillators. The presented experimental data, analytical and numerical results collectively are of broad interest. Potential impact of the work can be manifested in myriad of forms within disparate scientific endeavors ranging from physical to life sciences. The control of dynamics of nonlinear micromechanical systems represent an optimal sensing platform that allows for tailored functionalization of site specific properties opening many possibilities within nanobiotechnology, primarily in the construction of novel chemical and biological sensing systems based on non linear, high quality factor mechanical oscillators. Overall, the results of the paper are novel, the paper is well written and is of general interest with applicability and significant impact across disparate fields of sciences. I highly recommend the paper for publication in Nature after minor modifications outlined below.

Page 4: "The natural frequency of the principal in-plane flexural mode, $f_{in-plane}$, is found to be 61.4 kHz through electrostatic actuation and detection, with a linear damping rate $\gamma/2\pi = 0.50$ Hz, corresponding to a quality factor (Q) of 122680 (Fig. 1a)."

Throughout the paper, the authors report values for certain experimental quantities. In most instances, the values are reported without uncertainties as in the above quoted text from page 4 of the manuscript. For instance, reporting a value for the in-plane flexural mode can be accomplished in the following manner, either as an approximate value $f_{in-plane}$, is found to be approximately 61.4 kHz, same as $f_{in-plane} \approx 61.4$ kHz, or $f_{in-plane} \approx (61.4 \pm \text{some value})$ kHz. In the latter form, the author should comment on the source of the experimental measurement uncertainty. In the quoted text, the values of $\gamma/2\pi$ and Q need to be redefined in the manner outlined here. The authors should address this issue throughout the entirety of the paper.

Figure 1 a:

The red curve, appears to be a functional fit (mostlikely a Lorentzian), but is not defined in the figure or caption. The values of f_{res} are either approximate or have some measurement uncertainty, i.e. $f_{res} \approx 61.4$ kHz or $f_{res} = (61.4 \pm \text{some quantity})$ kHz. If the result is latter then the authors should

describe the source of error in an uncertainty analysis. Here, the error would be from the functional fit. Same comment for the reported values of the $\gamma/2\pi$ parameter in Fig 1a.

Comment concerning the y-axis, labeled "Amplitude (nm)": The experimental interferometric setup defined in the supplementary information Figure S1 references the work by P. Kanjanaboos et al in ref [S1]. Generally, with these optical interferometric systems, relative motion is measured. How was the absolute motion in "nanometers" experimentally measured in for the inplane mode of Fig 1a? More details to for the measurement and measurement calibrations of the motion should be included in the supplementary information.

Fig 1b and 1c caption: The value of 64.9 kHz has some spread. This should be defined as ≈ 64.9 kHz or $(64.9 \pm \text{some uncertainty})$ kHz, where the source of the uncertainty is then defined.

Fig 1c and 1d: The red curves in both figures should be defined.

Fig 2. Insets have values of $\gamma/2\pi$ that missing uncertainty. Similar to comments for Fig 1, these values are approximate defined using " \approx " or have an uncertainty. In this case the uncertainty would be determined from the functional fit used to extract $\gamma/2$. Similar comment for all values expressed in the Figure 2 caption.

Fig 2a, 2c, and 2e. The y-axis represents displacement in micrometers. It is not clear from the experimental setup (supplementary figure) how the magnitude of the absolute displacement is measured. It would be useful to include in the supplementary section the measurement methods, measurement calibration and measurement uncertainties for the displacement measurement of the ring down experiments.

Figure 2b and 2d: A comment regarding the color range within the two figures is necessary.

Figure 2f: What is the uncertainty in the t_{coherent} vs v_{ac} data points?

Figure S2a and S2b have frequency and quality factor values displayed. The quality factor value shows the uncertainty in the value whereas the frequency does not. It would be useful to state the frequency uncertainty and to comment on the source of the uncertainty – in this case the uncertainty would be from the Lorentzian functional fit.

Fig S1 shows that the device is placed into a vacuum chamber. At what pressure were the measurements made?

Supplementary section 1 makes mention that the MEMS devices were fabricated by MEMSCAP. It would be useful to include details of the fabrication process flow, schematic illustrations of the device design cross-sections, including the various materials (thin film layers) present and the resulting profiles following the release process.

REVIEWERS' COMMENTS:

Reviewer #1 (Remarks to the Author):

The authors have addressed all my comments in a way that I consider satisfactory. I do recommend publication.

Reviewer #2 (Remarks to the Author):

I find that the authors have carefully addressed the comments and criticism of the referees, and I now find it suitable for publication in Nature Communications.

Reviewer #3 (Remarks to the Author):

I unfortunately do not think the manuscript was much improved. I believe it should be made clear that this result builds on past efforts in the non-linear dynamics community, and that the only novelty here is that a known energy transfer and its mechanisms are used to sustain vibration. Again, I still think this is an exciting result. But the starting point in the introductory part should be the known unidirectional energy transfer and the associated beating. The authors are using these to investigate a bidirectional transfer ; it looks from the way they write the paper that they have brought to light the energy transfer in IR as well as the beating to explain their observation; this is a bit misleading. Why the new phenomenon had not been predicted before is also key. The authors write that this "might" be explained by different approximations in the work of Nayfeh and Mook (who considered the NL as a second order term for a reason). I strongly believe this point should be supported and made clear in the text.

About the impact: I agree that driving energy can be provided intermittently, but the total energy over a duty cycle has to remain identical, so I do not see the gain there.

The authors claim that MEMS/NEMS oscillators are limited by electrical noise. I believe this is wrong. Today, silicon oscillators are limited by temperature fluctuations. NEMS oscillators are already limited by their intrinsic frequency noise, as discussed in new ref 37. Moreover, the NL regime is carefully avoided in the operation of oscillators as this NL mixes $1/f$ noise from the circuitry (amplifier) to the resonant frequency, and this is well explained in the work of Rubiola too. I am wondering if this same NL would not demolish coherence of quantum systems. I agree the case of power outages is interesting though.

I believe proving one the claims would be required for publication in Nature Comms.

The easiest I can think of is to measure the Allan deviation of their system during the energy exchange with no external power and show that this frequency stability is limited by thermomechanical noise.

Reviewer #1 (Remarks to the Author):

The ms presents experimental observations of energy transfer between different “modes” in a nonlinear micro mechanical oscillator. The authors propose a description of the phenomenon in terms of nonlinear dynamics by introducing a Duffing oscillator coupled to a linear one.

The results are interesting. The experimental observation is novel and the phenomenon has potential applications in cases of practical interests.

We thank the referee for having reviewed the revised version of our manuscript and for noting the novelty and general importance of our work.

The experimental results are accurate and convincing, although a direct observation of the amplitude change of the high-frequency mode at the IR (Internal Resonance) condition is missing due to the limitations of the readout (optical) system employed by the authors.

The narrative of the paper is somehow misleading (probably beyond the author’s intentions). In fact this is presented as a way to avoid mechanical energy dissipation, while it is just a way to prolong self sustained oscillations by extracting energy from a different mode. Nevertheless a remarkable phenomenon. A careful rephrase of the introductory part of the ms could take care of the problem.

We have been very careful writing the introductory part of the manuscript to make evident that we are not claiming a mechanism to avoid mechanical energy dissipation in oscillators. Our findings provide a new strategy for *dissipation engineering* of MEMS resonators by capitalizing on the intrinsic nonlinear phenomena of these resonators. In order to satisfy the reviewer’s request, we have modified the introductory section of the manuscript to clarify our message.

A brief mention of the role of noise in the energy transfer process could widen the scope of the paper and potentially increase its interest.

The reviewer has correctly stated the importance of noise in the energy transfer process. Thermal and/or electronic noise acting on the micromechanical oscillator can be theoretically described as a random force, complementing Eq. (1) with a standard additive stochastic term with Gaussian statistics. The effect of this additional term, even for moderately large noise amplitudes, will be to blur the solutions of the presented phenomena showing in Figure 3.

This situation would be significant if the system were close to a critical (bifurcation) point, i.e. evolving in phase space close to an unstable state. If this were the case, even weak noise could drive the system outside internal resonance, perhaps avoiding the possibility of coherent energy transfer. For our system, we do not have either theoretical or experimental evidence that it could be in such a critical situation. Since we tried to balance the readability and yet thorough derivations, we intentionally excluded discussion about the effects of noise when describing our experimental findings.

We are currently planning to study the interplay between noise and mode coupling to identify the robustness of coherent energy transfer in nonlinear resonators.

In conclusion I find the work sound and potentially interesting for a wide audience. Worth publishing.

We thank again the reviewer for the positive comments and the recommendation to the editor.

Reviewer #2 (Remarks to the Author):

The paper by Chen et al. details ring-down experiments made on a MEMS resonator displaying a 3:1 internal resonance (IR). Their main finding is that free decay from initial conditions near the IR show a period of sustained coherent oscillations of the fundamental mode after the driving is switched off. They reproduce qualitatively the behavior using a simple model of linearly coupled oscillators.

The article is rather well written and for the most part clear, and deserves publication in some form. I have one major comment however (broken up into several below), that I think must be addressed before the article can be considered for publication.

We greatly appreciate the recommendations from the reviewer and the endorsement for publication. We will address the reviewer's comments below.

1. The two modes which are in resonance are in a 3:1 frequency ratio. As the authors point out very clearly on at least two places in the article, an internal resonance between the modes require a nonlinear coupling. Indeed, a coupling term in the Hamiltonian of the form $H_{12}=x_1^3x_2$ would for instance be able to do this.

1a) In the model they present, a linear coupling is used ($H_{12}=x_1x_2$). Can one argue that the excellent agreement between theory and experiment comes from a linearization of the nonlinear coupling at the amplitude near the IR? If so, this should be explained more clearly in the text.

We thank the reviewer for giving us the opportunity to clarify on this point.

In the manuscript, when presenting Eq. 1, we use the physical coordinates of x_1 and x_2 , which corresponds to the degrees of freedom of two coupled oscillators.

In order to discuss the Hamiltonians of the system, one needs to express the dynamics in modal coordinates, whose basis are orthogonal eigen-modes. In fact, one can transform from the linearly coupled model (expressed in physical coordinate, x_1, x_2) to the nonlinearly coupled normal form model (expressed in the modal coordinate q_1, q_2). When the system described by equation (1) is converted to the normal form, the coupling Hamiltonian term emerges as proportional to $q_1^3q_2$, as the reviewer correctly pointed out. We have added a new section in the SI, with a detailed discussion about the conversion between the degrees-of-freedom and the modal coordinates.

Since in our experimental setup, the directly measured quantities (e.g., capacitive currents from the sensing electrode) are proportional to the physical coordinates x_1 and x_2 , we choose to carry out the analysis solely in physical coordinates in the main text, in order to draw direct comparisons between the model and data.

The reviewer was correct in pointing out the confusion that arises when treating the system in physical coordinates, instead of modal coordinates. We have modified the manuscript accordingly, to clearly address this point.

1b) Towards the end of the supplementary, the authors argue that the coupling between the modes is linear. Indeed, it is explicitly written in the end of the paragraph after Eqs (S19a,b) that "..., therefore causing a linear coupling between the oscillations...". I do not understand how this linear coupling can account for the observed 3:1 internal resonance. This should be explained.

This is also related to question 1a above. Here the linear couplings are expressed in physical coordinate, which is equivalent to nonlinear couplings in modal coordinate. The observed 3:1

internal resonance can also be found from the eigen-modes, with small perturbations caused by the coupling. We have discussed the details in the new SI section.

1c) While the linearized coupling facilitates analytical estimates, I see no hindrance to simulating numerically the system with a nonlinear coupling for a true 3:1 internal resonance. I suggest the authors do so also in the supplementary in conjunction with figure S.10. This would be more convincing.

We believe the 1:1 internal resonance equations described in the manuscript are representative of a large variety of systems and thus, appealing to a general audience, beyond the experts in nonlinear dynamics.

We are currently working on a detailed model of a general $m:n$ internal resonance, both analytically and numerically, in order to identify the general trends and key parameters that control the energy transfer between modes.

1d) In all the modeling, all numbers are given in a dimensionless form. What is the relation between these numbers to actual physical values? (I am looking for actual numbers here). Can they be determined independently? Do the same parameters that reproduce the ringdown also reproduce the driven response? If one estimates the coupling constants J, J' do they agree with the ones in the simulations? How sensitive is the main result (stable amplitude of mode 1 after switching off the drive) to choice of parameters, i.e. is it a robust phenomenon or does it require careful parameter tuning?

The model we considered in equation (1) is a simplified one-dimensional model with a 1:1 coupling case, therefore it is difficult to trace the coupling coefficients J and J' directly back to physical values. In order to *quantitatively* reproduce the experimental results, a much more detailed model would be required (including materials properties, uncertainty in the fabrication process, higher order restoring forces, etc). The benefit of our model is that it captures the fundamental physics of our experiment with a simple equation as equation (1). Although more terms in this equation may give a better fit to the results, we seek, for clarity, to minimize the number of parameters involved.

The parameter values used in the simulations are chosen to best reproduce the experimental result *qualitatively*. However, if the parameters have clear physical origins, such as γ_1 and γ_2 , which correspond to the dissipations, we choose the parameter values close to the physical values: in this case, γ_1 and γ_2 are chosen to be on the order of 10^{-5} , that are quite close to the inverse quality factors. Moreover, the same parameters that reproduce the ring-down data also reproduce the driven response. The values of the parameters used in the simulations are listed in the manuscript.

1e) In relation to figure 4, the quantity γ_{ex} is introduced as the exchange rate. How does γ_{ex} relate to J, J' ? It seems it depends on drive power, why?

The energy exchange rate γ_{ex} is proportional to the product of J and J' , this can be seen directly when the system is transformed to modal coordinates, as shown in the equation S25, in the new section of the revised SI. In general, the energy exchange rate can be considered as the strength of the nonlinear coupling Hamiltonian (in modal coordinates). At a larger drive power, there is more

energy stored in the torsional mode during *steady state*, and we speculate this also contribute to a larger exchange rate. Currently we are working on a detailed model to quantify these effects.

We hope that based on our clarifications on modal coupling and modeling parameters and on the modifications introduced to the manuscript, the referee will recommend the revised manuscript for publication in *Nature Communications*.

Reviewer #3 (Remarks to the Author):

The result presented here is nice and simple: basically the authors have shown that the secondary mode of a resonator with 1:3 internal resonance transfers energy to the principal mode with a much faster rate than which of the considered modes, so that the principal mode sustains constant amplitude oscillations rather than decaying exponentially like a linear resonator. The experimental data looks very sound, results are clearly explained and well supported. This builds on a large body of work on inter-modal energy transfer mechanisms produced by the non-linear dynamics community in the last 50 years.

We would like to thank the referee for the recognition of our work and the constructive assessment. Moreover, we are thankful for the valuable comments regarding the current version of our manuscript.

I feel the authors might want to explain better why this is both surprising and important, starting with the title: aren't most of known energy transfers (parametric, sub/superharmonic, internal...) "coherent" and "directly observed"?

In spite of the fact that there have been demonstrations of energy transfer between different oscillatory systems, here we report the first ever observation of a mechanical system with prolonged, continuous oscillation, with constant amplitude *after* the external energy is removed. This phenomenon is counter-intuitive and, to our knowledge, has not been foreseen in any publication produced by the nonlinear dynamics community.

The authors write that concerning internal resonances, there is a "lack of definitive experimental results revealing the details of the energy transfer mechanism". I am not sure what they mean by that exactly. Could the authors relate this to both prior knowledge and their own result? If they have the sidebands (fig 4) in mind, hasn't this beating phenomenon already been shown by, for example, the work of Vakakis and co-authors (eg Nonlinear Targeted Energy Transfer in Mechanical and Structural Systems, Effect of 1:3 resonance on the steady-state dynamics of a forced strongly nonlinear oscillator with a linear light attachment AAM2014), or that of Manevitch and co-authors (ref23)? There is also much work performed by Pernot et al.

We found the works by Vakakis and co-authors highly relevant, and have added them to the manuscript references list. The works by Vakakis *et al.* and Pernot *et al.* are about reducing vibration energy using unidirectional energy transfer, facilitated by internal resonances. They also reported the beating behavior in internal resonances. However, they do not consider cases where energy transfer can be bi-directional, and used to sustain vibration in a mode - it's always about reducing vibration.

To the best of our knowledge, our result represents the first direct experimental observation of energy exchange between different vibrational modes that leads to constant amplitude and frequency of the fundamental mode, *without* external drive. There are very limited *experimental* results that directly show how energy is exchanged between modes. In our previous work (Antonio, Zanette, and López, *Nature Commun.* 3 (2012): 806.), the energy exchange results in the frequency stabilization of one vibrational mode, but no direct evidence of energy transfer is provided. Here, through the direct measurements of the energy decay (ring-down measurements), we have observed and confirmed the details of the coherent energy transfer.

Also, how does the results here compare to the work of Nayfeh and Mook, non-linear oscillations and in particular section 6.3.2 and figure 6-11 where both principal and secondary decay? (in this case, both modes have third order NL and the coupling is nonlinear too, can the authors comment on their linear coupling model?).

In the book by Nayfeh and Mook (Nonlinear Oscillations), which we have carefully examined during the development of our experiments, figure 6-11 shows the transient response of the modal amplitude in the presence of internal resonance. Both the modal amplitudes show oscillations (beating), but they *always* decay. The novelty of our case is that, there is no sign of amplitude decay of the primary mode during t_{coherent} , which we observed both theoretically and experimentally. The difference might arise from the different approximations. In Nayfeh and Mook's case, the authors only consider weak coupling, where the underlying model assumes the cubic nonlinearity is small compared to the linear components. Therefore, the nonlinear terms do not appear in the lowest order equation (6.3.3), but rather are treated as a higher-order approximation, as in equations (6.3.4) and (6.3.5). In contrast, we treat the cubic nonlinearity as the same order as the linear restoring force, instead of treating it as a first-order approximation.

More importantly, I am not sure I understand the potential impact of the result. In the authors' minds, how can this be used to "enable a new generation of autonomous devices"? Timescales of interest here are a few 100ms. They write this can be applied to oscillators limited by "electrical noise in the feedback circuit". How so?

The coherent time of ~ 300 ms demonstrated here can be a significant benchmark in fields such as quantum information, where decoherence of entangled quantum state due to thermal dissipation occurs on time scales of a few ms or shorter. The coupled equations we used to qualitatively explain our results are general and can be applied to a variety of quantum system, e.g., spins embedded in mechanical resonators, superconducting qubits inside a cavity, etc. We envision applications in the area of defense, where light and fast moving objects, e.g., projectiles and miniature drones, could incorporate these nonlinear resonators functioning as autonomous self-powered frequency sources (in GPS denied environments). Oscillators like the one described in the manuscript, can operate continuously even if the driving energy is provided intermittently, thus increasing the lifetime of the driving circuitry. In addition, the possibility to maintain a constant frequency and amplitude *without* external electrical power can be crucial in situations such as abrupt power outages or electromagnetic shockwaves. We hope that by reporting this novel finding to a broad audience, it will trigger genuine discussions in other related areas where the governing mechanisms are similar.

Currently, the performance of MEMS/NEMS oscillators is limited by the electrical feedback circuit – any noise originating from the sustaining feedback circuitry will be re-injected to the mechanical resonator and will deteriorate the frequency stability of the oscillator. During $t_{\text{coherence}}$ our oscillators can be operated without a feedback circuit, which will eliminate any noise not generated by the oscillator itself. The effects of feedback circuit noise on the performance of MEMS/NEMS oscillators are well known. Detailed studies describing these effects can be found in "Phase Noise and Frequency Stability in Oscillators" by Enrico Rubiola (Cambridge University Press).

What do they mean by "identifying the ultimate stability imposed by thermomechanical noise"? My understanding is that frequency fluctuations discussed in Ref 36 do not originate in the external power source.

As discussed in Ref 36, it is very difficult to remove the noise coming from electrical components in MEMS/NEMS as they act as part of the energy source, and these electrical components inevitably introduce noise. Here we present a new method where the oscillator can be operated and characterized without these external sources of noise, therefore opening the door to study the ultimate frequency stability as the only noise source is intrinsic to the mechanical oscillator.

As for non-equilibrium systems, I believe theories mostly refer to non-linear coupling (like coupling between flexural modes, eg Barnard et al, PNAS2012, Venstra et al APL2012), as opposed to the linear coupling considered here.

The reviewer is correct that most theories refer to nonlinear couplings. We note that although the coupling terms are linear, as shown in equation (1) involving degrees-of-freedom, when the same system is transformed to modal coordinates, the couplings become nonlinear. We have modified the manuscript to make this point clear, and added a new section in SI to outline the details of the coordinate transformation.

Again, I believe the result presented here is very nice, clearly shown and provides insight into the mechanism; I am just not sure the impact is sufficient for Nature Communications.

We thank again the reviewer for recognizing the quality of our work and hope that based on the reworked manuscript and our clarifications, the reviewer will reconsider his/her decision and recommend publication in *Nature Communications*.

Reviewer #4 (Remarks to the Author):

It has been a pleasure reviewing the paper entitled “Direct observation of coherent energy transfer in nonlinear micro-mechanical oscillators” by Chen et al. In recent years, continuing developments in micro and nanoelectromechanical systems (MEMS/NEMS) has fueled a renaissance in the field of miniaturized mechanical sensors and actuators. These devices have attained a great deal of consideration primarily since the practical significance of tailoring specific dynamical properties opens endless possibilities in the construction of a new class of sensors, actuators and new materials. Furthermore, MEMS and NEMS systems have produced an enormous impact in disparate scientific disciplines ranging from engineering to physical and life sciences, and furthermore have many commercial manifestations. Over the past few decades, these systems have shown to represent an ideal platform interface between the classical and quantum domains.

Nonlinear micromechanical systems represent structures that exhibit rich dynamical phenomena, and further have exquisite sensitivity near instabilities and allow for efficient configurational control of using external fields. The authors describe coherent energy transfer mechanisms in nonlinear micro-mechanical oscillators. The presented experimental data, analytical and numerical results collectively are of broad interest. Potential impact of the work can be manifested in myriad of forms within disparate scientific endeavors ranging from physical to life sciences. The control of dynamics of nonlinear micromechanical systems represent an optimal sensing platform that allows for tailored functionalization of site specific properties opening many possibilities within nanobiotechnology, primarily in the construction of novel chemical and biological sensing systems based on non linear, high quality factor mechanical oscillators. Overall, the results of the paper are novel, the paper is well written and is of general interest with applicability and significant impact across disparate fields of sciences. I highly recommend the paper for publication in Nature after minor modifications outlined below.

We would like to thank the referee for the enthusiastic evaluation of our work, and for the recommendation to the editor, and for the judgment on its impact both in the community of nano- and micro-mechanical oscillators and in other communities.

Page 4: “The natural frequency of the principal in-plane flexural mode, fin-plane, is found to be 61.4 kHz through electrostatic actuation and detection, with a linear damping rate $\gamma/2\pi = 0.50$ Hz, corresponding to a quality factor (Q) of 122680 (Fig. 1a).” Throughout the paper, the authors report values for certain experimental quantities. In most instances, the values are reported without uncertainties as in the above quoted text from page 4 of the manuscript. For instance, reporting a value for the in-plane flexural mode can be accomplished in the following manner, either as an approximate value fin-plane, is found to be approximately 61.4 kHz, same as fin-plane ≈ 61.4 kHz, or fin-plane $\approx (61.4 \pm \text{some value})$ kHz. In the latter form, the author should comment on the source of the experimental measurement uncertainty. In the quoted text, the values of $\gamma/2\pi$ and Q need to be redefined in the manner outlined here. The authors should address this issue throughout the entirety of the paper.

Following the reviewer’s suggestions, we have corrected these issues throughout the manuscript.

Figure 1 a:

The red curve, appears to be a functional fit (most likely a Lorentzian), but is not defined in the figure or caption. The values of f_{res} are either approximate or have some measurement uncertainty, i.e. $f_{res} \approx 61.4$ kHz or $f_{res} = (61.4 \pm \text{some quantity})$ kHz. If the result is latter then the authors should describe the source of error in an uncertainty analysis. Here, the error would be from the functional fit. Same comment for the reported values of the $\gamma/2\pi$ parameter in Fig 1a.

We have changed the equal signs to approximately equal signs, to reflect the uncertainties in the reported values. The uncertainties arise from the functional regression process.

Comment concerning the y-axis, labeled “Amplitude (nm)”: The experimental interferometric setup defined in the supplementary information Figure S1 references the work by P. Kanjanaboos et al in ref [S1]. Generally, with these optical interferometric systems, relative motion is measured. How was the absolute motion in “nanometers” experimentally measured in for the inplane mode of Fig 1a? More details to for the measurement and measurement calibrations of the motion should be included in the supplementary information.

The absolute displacement of the in-plane motion (shown in fig. 1a) is obtained from measuring the capacitive current with a lock-in amplifier, and the measured current is converted to mechanical displacement through known device geometry. We have added detailed discussion of the procedure is SI, section I. We only use the optical interferometry setup to measure the thermomechanical noise spectrum of the torsional mode, and we performed separated calibration steps to obtain the responsivity of the interferometry setup. We have also added such discussion in the SI, section I.

Fig 1b and 1c caption: The value of 64.9 kHz has some spread. This should be defined as ≈ 64.9 kHz or $(64.9 \pm \text{some uncertainty})$ kHz, where the source of the uncertainty is then defined.

We have made the changes accordingly. The spread around 64.9 kHz is due to measurement noise and data fitting processes.

Fig 1c and 1d: The red curves in both figures should be defined.

We have added the definitions of the respective red curves in the caption.

Fig 2. Insets have values of $\gamma/2\pi$ that missing uncertainty. Similar to comments for Fig 1, these values are approximate defined using “ \approx ” or have an uncertainty. In this case the uncertainty would be determined from the functional fit used to extract $\gamma/2$. Similar comment for all values expressed in the Figure 2 caption.

We have changed the equal signs to approximately equal signs to reflect the uncertainties. As the reviewer correctly pointed out, the uncertainties come from the fittings to an exponential decay function.

Fig 2a, 2c, and 2e. The y-axis represents displacement in micrometers. It is not clear from the experimental setup (supplementary figure) how the magnitude of the absolute displacement is

measured. It would be useful to include in the supplementary section the measurement methods, measurement calibration and measurement uncertainties for the displacement measurement of the ring down experiments.

The displacement of the in-plane motion (expressed in micrometers) is converted from the measured voltage acquired by the oscilloscope. We have added the detailed procedures in SI, section I. The measurement calibration is done by comparing the measured voltage with the images acquired directly from a camera. The details are described in the supplementary information of Ref. 32, section 11.

Figure 2b and 2d: A comment regarding the color range within the two figures is necessary.

Figure 2b and 2d show the temporal fast Fourier transform (FFT) for the time domain data shown in figure 2a and 2c, respectively. The color scale in figure 2b and 2d represents the power of the FFT at each FFT frequency and time, expressed in logarithmical scale. We have added the necessary color bars to reflect the power range.

Figure 2f: What is the uncertainty in the t_{coherent} vs V_{ac} data points?

The uncertainty of t_{coherent} arises from data extraction process. After re-examining the data, it turns out the values of the uncertainty are smaller than the sizes of the plotted data points, therefore we will not show the error bars in this case. We have added this result in the figure caption.

Figure S2a and S2b have frequency and quality factor values displayed. The quality factor value shows the uncertainty in the value whereas the frequency does not. It would be useful to state the frequency uncertainty and to comment on the source of the uncertainty – in this case the uncertainty would be from the Lorentzian functional fit.

We thank the reviewer for identifying this oversight, and we have added the uncertainties in the resonant frequencies. As the reviewer pointed out, the uncertainties are from the Lorentzian fits.

Fig S1 shows that the device is placed into a vacuum chamber. At what pressure were the measurements made?

The pressure is estimated to be 3.0×10^{-5} Torr. The vacuum gauge is placed at the outlet of the vacuum chamber; therefore, the pressure inside the chamber is expected to be slightly higher. We have added this information to fig. S1 and SI.

Supplementary section 1 makes mention that the MEMS devices were fabricated by MEMSCAP. It would be useful to include details of the fabrication process flow, schematic illustrations of the device design cross-sections, including the various materials (thin film layers) present and the resulting profiles following the release process.

The devices are fabricated by MEMSCAP, with standard SOIMUMPs process (Silicon-On-Insulator, Multi-User MEMS Processes). We have added the weblink that outlines the detailed information of the fabrication process flow in the SI.

Reviewer #3 (Remarks to the Author):

I unfortunately do not think the manuscript was much improved.

I believe it should be made clear that this result builds on past efforts in the non-linear dynamics community, and that the only novelty here is that a known energy transfer and its mechanisms are used to sustain vibration.

We believe that our manuscript is very clear about the phenomena we are reporting. To the best of our knowledge, our result represents the first direct experimental observation of energy exchange between different vibrational modes that leads to *constant amplitude and frequency* of the fundamental mode, without external drive. This phenomenon is counter-intuitive and, to our knowledge, has not been foreseen in any publication produced by the nonlinear dynamics community in past years.

Again, I still think this is an exciting result. But the starting point in the introductory part should be the known unidirectional energy transfer and the associated beating. The authors are using these to investigate a bidirectional transfer; it looks from the way they write the paper that they have brought to light the energy transfer in IR as well as the beating to explain their observation; this is a bit misleading.

As we describe in our manuscript and in our previous response to this reviewer, we are reporting a new phenomena originated from a bidirectional transfer of energy between coupled mechanical modes. To justify our conclusions, we have listed a coupled of well-known books on the subject and quite a few related references.

Why the new phenomenon had not been predicted before is also key. The authors write that this "might" be explained by different approximations in the work of Nayfeh and Mook (who considered the NL as a second order term for a reason). I strongly believe this point should be supported and made clear in the text.

We cannot speculate about why something was not discovered before it was. Such a request imposes an almost impossible task that does not improve the explanation of the observations nor does it improve the understanding of the phenomenon. We have already addressed how our work is significantly different than the vast majority of previous work. In Nayfeh and Mook's case, the authors only consider weak coupling, where the underlying model assumes the cubic nonlinearity is small compared to the linear components. In our experiments, our detuning range is considerably large (~ 10%). We speculate that since this nonlinear regime is difficult to achieve experimentally, there has not been many theoretical studies of mode coupling with large Duffing coefficients. We predict that our results will renew interest in studying this deep nonlinear regime and thus, novel phenomenon would be predicted.

About the impact: I agree that driving energy can be provided intermittently, but the total energy over a duty cycle has to remain identical, so I do not see the gain there.

Oscillators like the one described in the manuscript, can operate continuously even if the driving energy is provided intermittently, thus increasing the lifetime of the driving circuitry. The gain is in the lifetime of the electronic used for driving, not in the MEMS oscillator.

The authors claim that MEMS/NEMS oscillators are limited by electrical noise. I believe this is wrong. Today, silicon oscillators are limited by temperature fluctuations. NEMS oscillators are already limited by their intrinsic frequency noise, as discussed in new ref 37.

In our manuscript we specifically wrote: “The dissipation engineering concept presented in this work could be applied to a wide range of MEMS and NEMS oscillators whose performance is limited by the electrical noise in the feedback circuit”. As a consequence, for the wide range of MEMS and NEMS oscillators whose performance is limited by electrical noise in the feedback circuit, the mechanism we are reporting here can provide a path to improve their performance.

Silicon oscillators are limited by a large number of parameters, not only by temperature fluctuations. A portion of the noise is internal, resulting from the resonator itself, due to fluctuations of the natural frequency, stiffness and dissipation, or due to thermo-mechanical noise causing Brownian motion of the resonator. Noise also originates from the feedback loop in the saturation of the feedback signal. In addition to these well-studied sources of noise, reference 32 (the same one as the ref 37 that the reviewer is referring to) claims that an additional source of noise -- whose origin is not understood -- limits the ultimate frequency stabilization achievable with silicon oscillators.

Moreover, the NL regime is carefully avoided in the operation of oscillators as this NL mixes $1/f$ noise from the circuitry (amplifier) to the resonant frequency, and this is well explained in the work of Rubiola too. I am wondering if this same NL would not demolish coherence of quantum systems. I agree the case of power outages is interesting though.

I believe proving one the claims would be required for publication in Nature Comms. The easiest I can think of is to measure the Allan deviation of their system during the energy exchange with no external power and show that this frequency stability is limited by thermomechanical noise.

We appreciate the suggestion and are planning to perform a similar experiment in the near future to identify the origin of the additional source of noise reported in reference 32 (the same one as the ref 37 that the reviewer is referring to).